

# A new metaphor-less simple algorithm based on Rao algorithms: a Fully Informed Search Algorithm (FISA)

Mojtaba Ghasemi[1,*], Abolfazl Rahimnejad[2,*], Ebrahim Akbari[3], Ravipudi Venkata Rao[4], Pavel Trojovský[3], Eva Trojovská[3] and Stephen Andrew Gadsden[2]

[1] Department of Electronics and Electrical Engineering, Shiraz University of Technology, Shiraz, Iran
[2] Department of Mechanical Engineering, McMaster University, Hamilton, Canada
[3] Department of Mathematics, Faculty of Science, University of Hradec Králové, Hradec Králové, Czech Republic
[4] Department of Mechanical Engineering, Sardar Vallabhbhai National Institute of Technology, Ichchanath, Surat, Gujarat, India
[*] These authors contributed equally to this work.

Corresponding author
Ebrahim Akbari,
ebrahimakbary@gmail.com,
ebrahim.akbari@uhk.cz

## ABSTRACT

Many important engineering optimization problems require a strong and simple optimization algorithm to achieve the best solutions. In 2020, Rao introduced three non-parametric algorithms, known as Rao algorithms, which have garnered significant attention from researchers worldwide due to their simplicity and effectiveness in solving optimization problems. In our simulation studies, we have developed a new version of the Rao algorithm called the Fully Informed Search Algorithm (FISA), which demonstrates acceptable performance in optimizing real-world problems while maintaining the simplicity and non-parametric nature of the original algorithms. We evaluate the effectiveness of the suggested FISA approach by applying it to optimize the shifted benchmark functions, such as those provided in CEC 2005 and CEC 2014, and by using it to design mechanical system components. We compare the results of FISA to those obtained using the original RAO method. The outcomes obtained indicate the efficacy of the proposed new algorithm, FISA, in achieving optimized solutions for the aforementioned problems. The MATLAB Codes of FISA are publicly available at https://github.com/ebrahimakbary/FISA.

## INTRODUCTION

The objective of maximizing profits or minimizing losses is a crucial concern in several fields, including engineering. In brief, an optimization problem refers to the situation where the aim is to maximize or minimize a function. With the development of technology, optimization problems have become increasingly complex and abundant across a wide range of scientific fields. The complexity and interdependence of modern engineering systems and problems necessitate the selection of the best optimization method to solve

them. Metaheuristic algorithms are among the strongest, simplest, and most commonly used optimization methods in recent years (*Gogna & Tayal, 2013*; *Zervoudakis & Tsafarakis, 2020*).

In general, a mathematical model in the optimization process has three main parts: objective function, design variables, and problem and system constraints. Design or decision variables are the independent variables that must be optimally determined and are denoted by the D-dimensional vector $X$. According to the problem's nature, they can have a combination of several types of discrete and continuous decision variables. The objective function, also called the cost function, is a function of the decision variables that should be minimized or maximized. The goal of solving optimization problems is to obtain an acceptable solution that minimizes/maximizes objective function while satisfying the problem constraints. The constraints are the same as the physical and design constraints of the problem that must be satisfied in the optimization process so that a practical optimal solution can be obtained (*Gogna & Tayal, 2013*).

Optimization problems can be broadly categorized into two types: unconstrained optimization problems and constrained optimization problems. In the latter case, the design space is limited by one or more constraints, which can take the form of equality or inequality equations. These constraints determine the acceptable region in the design space where the optimal solution must be found.

## Optimization algorithms

Optimization algorithms can be broadly classified into two types: exact methods and approximate methods. Exact methods are capable of guaranteeing an optimal solution, but they may require significant computational resources and time. In contrast, approximate methods focus on finding good solutions in a reasonable amount of time. Heuristic algorithms are a popular type of approximate methods that are designed to quickly generate high-quality solutions to a wide range of problems. The effectiveness of heuristic algorithms depends on the nature of the problem being solved (*Gogna & Tayal, 2013*).

## Metaheuristic methods

Metaheuristics refer to methods that guide the search process and are often inspired by nature. Unlike heuristic algorithms, due to their problem-independent nature, these algorithms can be utilized to optimize a diverse range of problems. These methods are among the most important and promising research in the optimization domain.

The general principles of metaheuristic methods are as follows:

- Employing a given number of repetitive efforts
- Employing one or more agents (particles, neurons, ants, chromosomes, *etc.*)
- Operation (in multi-factor mode with a cooperation-competition mechanism)
- Creating methods of self-change and self-transformation

Nature has two great tactics:
1. Rewarding strong personal characteristics and punishing weaker ones.
2. Introducing random mutations, which can lead to the birth of new individuals.

Recently, many optimization techniques have been proposed that operate on the basis of natural behaviors and social behaviors. In the initialization stage, these algorithms randomly generate solutions and in the later stages, they rely on natural processes to produce better answers.

Some of the popular and widely used types of metaheuristics are Particle swarm optimization algorithm (PSO) (*Kennedy & Eberhart, 1995*), differential evolution (DE) (*Storn & Price, 1995*), genetic algorithm (GA) (*Whitley, 1994*), firefly algorithm (*Yang, 2009*), ant colony optimization (*Dorigo & Di Caro*), bat algorithm (*Yang, 2010*), teaching–learning-based optimization (TLBO) (*Rao, Savsani & Vakharia, 2011*), grey wolf optimizer (GWO) (*Mirjalili, Mirjalili & Lewis, 2014*), artificial bee colony algorithm (ABC) (*Karaboga & Basturk, 2007*), imperialist competitive algorithm (ICA) (*Atashpaz-Gargari & Lucas, 2007*), moth-flame optimization algorithm (MFO) (*Mirjalili, 2015*), gravitational search algorithm (GSA) (*Rashedi, Nezamabadi-pour & Saryazdi, 2009*), shuffled frog-leaping algorithm (SFLA) (*Eusuff, Lansey & Pasha, 2006*), whale optimization algorithm (WOA) (*Mirjalili & Lewis, 2016*), *etc.*

Now the question is why all these new optimization algorithms, either modified or combined, are needed. The main reason for this is the inability to determine with certainty which optimization or metaheuristic algorithm is appropriate for resolving a problem, and only through the comparison of the outcomes can it be asserted which algorithm provides a superior approach. In addition, based on (*Mirjalili, Mirjalili & Lewis, 2014*), an algorithm may perform well for some groups of functions but not for some other groups. Therefore, the motivation to modify algorithms or introduce new algorithms has been very high in recent years (*Mirjalili & Lewis, 2016*).

In 2020, *Rao (2020)* suggested three effective and powerful straightforward algorithms for optimization problems without the use of metaphors. These methods use the most and least optimal solutions in each iteration and the casual interrelations between possible solutions. Additionally, these methods need no control parameter other than the population size and the number of iterations.

Rao algorithms have been successfully used by researchers in the short time since they have been introduced, some of which include the engineering design optimization (*Rao & Pawar, 2020a*; *Rao & Pawar, 2020b*; *Rao & Pawar, 2022*; *Rao & Pawar, 2023*), weight optimization of reinforced concrete cantilever retaining wall (*Kalemci & Ikizler, 2020*), cropping pattern under a constrained environment (*Kumar & Yadav, 2019*), construction scheduling (*Yılmaz & Dede, 2023*), photovoltaic cell parameter estimation (*Premkumar et al., 2020*; *Wang et al., 2020*), optimization of energy systems (*Rao et al., 2022*), and optimal power flow (*Sahay, Upputuri & Kumar, 2023*).

A significant proportion of optimization problems encountered in practical applications contain shifted functions, for which the performance of Rao algorithms may not be much optimal, as demonstrated in the simulation section. This article introduces a new algorithm, called Fully Informed Search Algorithm (FISA), which is based on Rao algorithms and address this drawback. FISA not only outperforms the original Rao algorithms in optimizing shifted functions but also preserves their simplicity and requires no control parameters. The suggested algorithm's performance is evaluated by optimizing benchmark problems with

shifted functions and real-world problems. The results demonstrate that FISA outperforms not only the original Rao algorithms but also other state-of-the-art methods, indicating its superior performance.

The article continues in four sections: the formulation of the suggested and Rao algorithms is presented in the following section. Reporting the outcomes of simulations and presenting and discussing the findings are done in next section. Lastly, the conclusions are presented.

## PROPOSED ALGORITHM

### Rao algorithms

The basic formulations of Rao algorithms, which are very simple algorithms without control parameters, rely on the difference vectors obtained by subtracting the position (location) of the worst individual from the location of the finest individual in the present iteration. This ensures that the population always moves towards a better solution. These algorithms consist of three distinct movements (position update) vectors for updating the position which are defined as follows (*Rao, 2020*):

Rao-1 algorithm:

$$X_{i,j}^{\text{new}} = X_{i,j}^{\text{Iter}} + r_{1,j}\left(X_{\text{best},j}^{\text{Iter}} - X_{\text{worst},j}^{Iter}\right). \tag{1}$$

Rao-2 algorithm:

$$\begin{cases} \text{if } f\left(X_i^{\text{Iter}}\right) < f\left(X_k^{\text{Iter}}\right) \\ X_{i,j}^{\text{new}} = X_{i,j}^{\text{Iter}} + r_{1,j}\left(X_{\text{best},j}^{\text{Iter}} - X_{\text{worst},j}^{\text{Iter}}\right) + r_{2,j}\left(\left|X_{i,j}^{\text{Iter}}\right| - \left|X_{k,j}^{\text{Iter}}\right|\right); \\ \text{else} \\ X_{i,j}^{\text{new}} = X_{i,j}^{\text{Iter}} + r_{1,j}\left(X_{\text{best},j}^{\text{Iter}} - X_{\text{worst},j}^{\text{Iter}}\right) + r_{2,j}\left(\left|X_{k,j}^{\text{Iter}}\right| - \left|X_{i,j}^{\text{Iter}}\right|\right). \end{cases} \tag{2}$$

Rao-3 algorithm:

$$\begin{cases} \text{if } f\left(X_i^{\text{Iter}}\right) < f\left(X_k^{\text{Iter}}\right) \\ X_{i,j}^{\text{new}} = X_{i,j}^{\text{Iter}} + r_{1,j}\left(X_{\text{best},j}^{\text{Iter}} - \left|X_{\text{worst},j}^{\text{Iter}}\right|\right) + r_{2,j}\left(\left|X_{i,j}^{\text{Iter}}\right| - X_{k,j}^{\text{Iter}}\right); \\ \text{else} \\ X_{i,j}^{\text{new}} = X_{i,j}^{\text{Iter}} + r_{1,j}\left(X_{\text{best},j}^{\text{Iter}} - \left|X_{\text{worst},j}^{\text{Iter}}\right|\right) + r_{2,j}\left(\left|X_{k,j}^{\text{Iter}}\right| - X_{i,j}^{\text{Iter}}\right). \end{cases} \tag{3}$$

In the above equations, $X_i^{\text{Iter}}$ represents the $i$th solution's location in the present iteration Iter; $j$ (changing from 1 to $D$) represents the $j$th dimension of each solution; $X_{\text{best}}^{\text{Iter}}$ and $X_{\text{worst}}^{\text{Iter}}$ represent the position of the highest and lowest performing members of the population during the present iteration, in that order; $r_1$ and $r_2$ are two randomly selected values between 0 and 1 with the dimension of $D$; $X_k^{\text{Iter}}$ represents the position of the $k$th solution, which is indiscriminately chosen; and $f$ (.) represents the numerical output of the function being optimized of the corresponding solution in the present iteration. The location of the $i$th solution in the next iteration is obtained using Eq. (4):

$$\begin{cases} X_i^{\text{Iter}+1} = X_i^{\text{new}} \text{ if } f\left(X_i^{\text{new}}\right) \le f\left(X_i^{\text{Iter}}\right) \\ X_i^{\text{Iter}+1} = X_i^{\text{Iter}} \text{ else}. \end{cases} \tag{4}$$

### The proposed Fully Informed Search Algorithm (FISA)

The performance of Rao algorithms in optimizing shifted functions, which may be the case for many real-world problems, may be suboptimal. This will be demonstrated in the simulation section. Therefore, we propose a new algorithm, the Fully Informed Search Algorithm (FISA), which is based on Rao algorithms and is designed to address this issue. FISA improves the optimization of shifted functions while retaining the simplicity and absence of control parameters of the original algorithms. Similar to Rao algorithms, FISA moves the population towards better solutions. In summary, FISA can be summarized as follows:

$$X_{i,j}^{\text{new}} = X_{i,j}^{\text{Iter}} + r_{1,j}\left(MX_{\text{best},j}^{\text{Iter}} - X_{i,j}^{\text{Iter}}\right) + r_{2,j}\left(X_{i,j}^{\text{Iter}} - MX_{\text{worst},j}^{\text{Iter}}\right). \tag{5}$$

In fact, in FISA, each member moves away from the mean position of the individuals within the population that have worse fitness values and approaches the mean position of the individuals that have better fitness values than the associated member. Then, the position of each member is updated using Eq. (4). In Eq. (5), the values of $MX_{\text{best},j}^{\text{Iter}}$ and $MX_{\text{worst},j}^{\text{Iter}}$ in each iteration are calculated using Eqs. (6) and (7), respectively:

$$MX_{\text{best}}^{\text{Iter}} = \frac{X_{\text{best}}^{\text{Iter}} + \sum_{l \in Bi} X_l^{\text{Iter}}}{\text{length}(Bi) + 1} \tag{6}$$

$$MX_{\text{worst}}^{\text{Iter}} = \frac{X_{\text{worst}}^{\text{Iter}} + \sum_{l \in Wi} X_l^{\text{Iter}}}{\text{length}(Wi) + 1} \tag{7}$$

where $B_i$ and $W_i$ are the set of population members that have a better and worse fitness value than the $i^{th}$ member in iteration *Iter*, respectively, and *length*(.) represents the count of the individuals in the set.

The flowchart of FISA is shown in Fig. 1.

## NUMERICAL RESULTS OF FISA FOR SOLVING BENCHMARK TEST FUNCTIONS

### FISA for solving CEC2005 problems

To assess the effectiveness of the suggested algorithm in comparison to the original Rao algorithms, we have chosen 14 real-world shifted functions with 30 dimensions (including unimodal, multimodal, and expanded multimodal functions), numbered in the order introduced in CEC 2005 (*Liu et al., 2013*), whose data were extracted from *Suganthan et al. (2005)*. These functions have been successfully utilized in many articles (*Ghasemi, Aghaei & Hadipour, 2017*; *Birogul, 2019*; *Ghasemi et al., 2019*; *Ghasemi et al., 2022a*; *Ghasemi et al., 2022b*; *Ghasemi et al., 2023*; *Akbari, Rahimnejad & Gadsden, 2021*; *Premkumar et al., 2021*; *Zou et al., 2022*). The total count of function evaluations (NFE) during the execution of each algorithm is considered 300,000 based on *Liu et al. (2013)*; accordingly, the number of population members of each algorithm during the optimization is 30; therefore, the convergence curves, in this study, have been extracted for 10,000 iterations. Furthermore, to acquire the optimal solution for each function, each algorithm has been executed

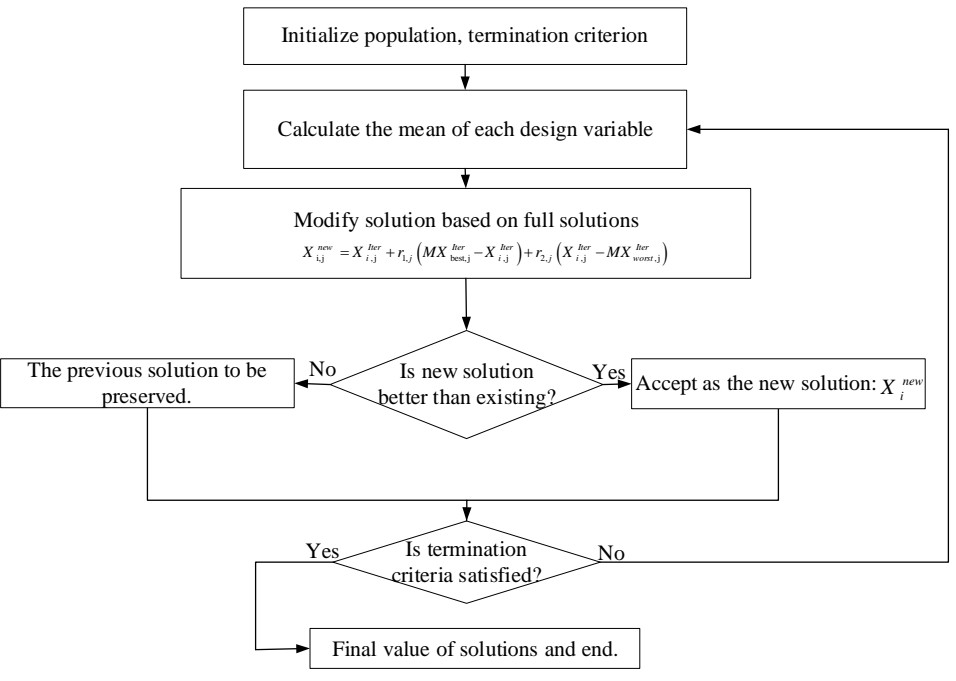

**Figure 1 Flowchart of FISA.**

independently for 25 runs. A summary of the results comprising the average value, standard deviation, and rank for each algorithm amongst the investigated algorithms is given in Table 1. In this table, $N_b$ and $N_w$ represent the total instances where the corresponding algorithm has the best or the worst result among the studied algorithms, respectively. $M_R$ also represents the mean ranking of each algorithm for 14 test functions.

After a concise investigation of Table 1, it can be observed that the FISA algorithm noticeably outperformed the original Rao algorithms, particularly for functions F2, F4, and F6. The proposed algorithm surpassed the Rao-2 and Rao-3 algorithms for all investigated functions, except for the F8 test function where it achieved the same performance as the Rao algorithms. Although the FISA algorithm showed slightly lower performance than the Rao-1 algorithm for test functions F1, F7, and F9, these results were close, and FISA outperformed the Rao-1 algorithm for the remaining 10 test functions. These findings indicate that the proposed algorithm has a strong capability to achieve optimal solutions for practical problems. Additionally, the convergence behaviors of different algorithms for solving the selected functions are presented in Fig. 2, which provides evidence of the superior convergence behavior of the suggested algorithm.

## FISA for solving CEC2014 problems

In the second part of demonstrating the efficacy of the suggested method, FISA, in comparison with the RAO algorithms, 30 test functions from CEC 2014 Test Functions, with dimension 30 are selected (*Askari & Younas, 2021*; *Suganthan et al., 2005*; *Liu & Nishi, 2022*; *Meng et al., 2022*; *Band et al., 2022*), whose data were extracted from (*Suganthan et*

**Table 1** The optimal results obtained by Rao algorithms and FISA on the 30-D real-parameter test functions of CEC 2005.

| F | Rao-1 Mean Std Dev Rank | Rao-2 Mean Std Dev Rank | Rao-3 Mean Std Dev Rank | FISA Mean Std Dev Rank |
|---|---|---|---|---|
| F1 | 2.88E−29 7.64E−29 1 | 2.39E−05 3.43E−05 3 | 4.11E+03 5.53E+02 4 | 3.37E−27 5.54E−27 2 |
| F2 | 3.32E−07 8.76E−07 2 | 7.53E+03 2.52E+03 3 | 1.51E+04 2.91E+03 4 | 2.04E−26 3.89E−27 1 |
| F3 | 1.61E+07 8.89E+06 2 | 7.02E+07 2.19E+07 3 | 8.41E+07 2.36E+07 4 | 5.83E+06 3.84E+06 1 |
| F4 | 2.18E+02 5.09E+02 4 | 2.09E+04 8.84E+03 2 | 2.11E+04 8.43E+03 3 | 4.01E−05 9.75E−05 1 |
| F5 | 3.63E+03 1.70E+03 3 | 3.35E+03 2.44E+03 2 | 5.56E+03 2.07E+03 4 | 2.34E+03 9.42E+03 1 |
| F6 | 2.19E+01 3.54E+01 2 | 5.55E+06 1.47E+07 3 | 1.09E+08 2.51E+07 4 | 4.01E+00 4.96E+07 1 |
| F7 | 1.98E−02 1.36E−02 1 | 2.64E−01 2.50E−01 3 | 3.03E+02 5.19E+01 4 | 2.88E−02 1.32E−02 2 |
| F8 | 2.09E+01 5.39E−02 1 | 2.09E+01 3.42E−02 1 | 2.09E+01 6.51E−02 1 | 2.09E+01 5.69E−02 1 |
| F9 | 1.63E+02 5.71E+01 1 | 2.02E+02 1.44E+01 3 | 2.33E+02 1.21E+01 4 | 1.91E+02 2.00E+01 2 |
| F10 | 2.28E+02 2.15E+01 2 | 2.34E+02 1.48E+01 3 | 3.03E+02 2.60E+01 4 | 1.78E+02 1.99E+01 1 |
| F11 | 4.02E+01 1.25E+00 4 | 3.93E+01 5.42E−01 3 | 3.89E+01 1.11E+00 2 | 3.77E+01 1.65E+00 1 |
| F12 | 6.11E+04 5.85E+04 2 | 1.93E+05 1.60E+05 4 | 1.62E+05 6.39E+04 3 | 5.28E+04 5.05E+04 1 |

**Table 1** (*continued*)

| F | Rao-1 Mean Std Dev Rank | Rao-2 Mean Std Dev Rank | Rao-3 Mean Std Dev Rank | FISA Mean Std Dev Rank |
|---|---|---|---|---|
| | 1.70E +01 | 1.78E +01 | 3.38E +01 | 1.50E +01 |
| F13 | 8.39E −01 | 1.61E +00 | 3.59E +00 | 1.23E +00 |
| | 2 | 3 | 4 | 1 |
| | 1.33E +01 | 1.34E +01 | 1.34E +01 | 1.27E +01 |
| F14 | 2.29E −01 | 2.08E −01 | 1.16E −01 | 2.02E −01 |
| | 2 | 3 | 3 | 1 |
| Nb/Nw/MR | 4/2/2.071 | 1/2/2.786 | 1/10/3.429 | 11/0/1.214 |

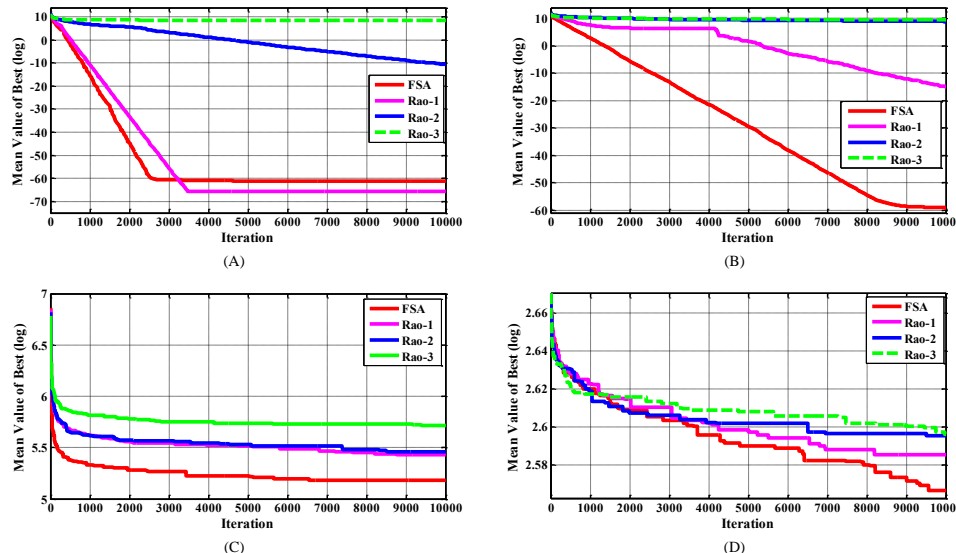

**Figure 2** **The convergence characteristics of FISA and the Rao algorithms for some selected CEC2005 benchmark functions.** (A) F1 test function. (B) F2 test function. (C) F10 test function. (D) F14 test function.

*al., 2005*). The optimal value for all functions is 0. The population number was selected as 30 and the stopping criterion was selected as 10,000 iterations for all algorithms; so that the NFE for each algorithm is equal to 300,000. The experiment is conducted by running each algorithm independently 25 times for every function. The summary of the results comprising the Mean value, standard deviation, and best value for each of the investigated algorithms is given in Table 2.

After reviewing the results presented in Table 2, taking into account the average value, standard deviation, and also the best optimal value of 25 runs, we can see that the suggested algorithm has a significant advantage over the Rao algorithms. The last row of the table shows the number of test functions in which each algorithm achieved their best solution, denoted as Nb. It is evident that the proposed method outperformed the other algorithms by obtaining the best solution in 23 out of 30 test functions. This indicates the strength and

efficiency of the suggested technique as a new optimization algorithm. The convergence characteristics of algorithms for some selected test functions are displayed in Fig. 3.

## NUMERICAL RESULTS OF FISA IN SOLVING ENGINEERING PROBLEMS

To demonstrate the effectiveness and optimization efficiency of FISA, three widely recognized engineering problems, namely optimal design of a pressure vessel, tension/compression spring, and welded beam, have been selected. Then, we perform the optimization operation for these optimization problems under the same conditions for all algorithms. The number of populations for each algorithm is chosen as 60, and the number of iterations of the algorithms for each run is chosen as 2000. In addition, each optimization operation is performed in 30 separate runs for each problem, using all the parameters as suggested by the respective algorithm designers in their original publications.

### Pressure vessel optimal design

The goal of the problem is to minimize the overall costs of a pressure vessel, comprising material, forming, and welding expenses. As depicted in Fig. 4, this problem involves four design variables: shell thickness (denoted as $x_1$ or $T_s$), head thickness (denoted as $x_2$ or $T_h$), inner radius (denoted as $x_3$ or $R$), and cylindrical section length (denoted as $x_4$ or $L$). While $x_3$ and $x_4$ are continuous variables, $x_1$ and $x_2$ are discrete variables represented as integer multiples of 0.0625 in.

The problem's objective function is nonlinear and it has both a linear and a nonlinear inequality constraint, which are illustrated below (*Askarzadeh, 2016*):

Minimize:

$$f(X) = 0.6224x_1x_3x_4 + 1.7781x_2x_3^2 + 3.1661x_1^2x_4 + 19.84x_1^2x_3 \tag{8}$$

subject to:

$$g_1(X) = -x_1 + 0.0193x_3 \leq 0, \tag{9}$$

$$g_2(X) = -x_2 + 0.00954x_3 \leq 0, \tag{10}$$

$$g_3(X) = -\pi x_3^2 x_4 - \frac{4}{3}\pi x_3^3 + 1,296,000 \leq 0, \tag{11}$$

$$g_4(X) = x_4 - 240 \leq 0, \tag{12}$$

$0 \leq x_i \leq 100, i = 1, 2$
$10 \leq x_i \leq 200, i = 3, 4.$

Table 3 compares the outcomes achieved by the proposed algorithm for the problem and other widely used standard algorithms, including quantum-inspired PSO (QPSO)

**Table 2  Mean statistical results of the optimization algorithms.** The optimal results obtained by Rao algorithms and FISA on the 30-D real-parameter test functions of CEC 2014.

| Number Functions | Mean | | | | Std. | | | | Best | | | |
|---|---|---|---|---|---|---|---|---|---|---|---|---|
| | Rao-1 | Rao-2 | Rao-3 | FISA | Rao-1 | Rao-2 | Rao-3 | FISA | Rao-1 | Rao-2 | Rao-3 | FISA |
| F1 | 9582162 | 50437582 | 81428816 | 2800719 | 3079516 | 20123351 | 39651820 | 2150031 | 6946796 | 14997575 | 29955287 | 628863 |
| F2 | 120.89 | 659713 | 6137293771 | 0.2668 | 112.9 | 848640 | 2928707621 | 0.2912 | 0.4824 | 29241 | 3010346193 | 0.008785 |
| F3 | 4485 | 40811 | 38797 | 31.1 | 2228.7 | 10093 | 9319 | 31.85 | 1526 | 24146 | 23581 | 0.4269 |
| F4 | 61.73 | 133.42 | 342.18 | 24.415 | 51.53 | 39.751 | 69.579 | 31.2 | 1.21 | 64.89 | 270.3 | 1.379e−05 |
| F5 | 20.90 | 20.87 | 20.93 | 20.94 | 0.06756 | 0.0954 | 0.0494 | 0.06964 | 20.79 | 20.65 | 20.85 | 20.77 |
| F6 | 15.33 | 30.14 | 30.72 | 9.535 | 7.284 | 5.586 | 4.372 | 2.443 | 6.486 | 15.29 | 22.86 | 6.25 |
| F7 | 0.005914 | 0.0994 | 15.59 | 0.00689 | 0.005473 | 0.1606 | 5.7095 | 0.00919 | 3.41e−13 | 2.27e−13 | 10.26 | 3.41e−13 |
| F8 | 171.18 | 194.74 | 194.73 | 183.55 | 45.848 | 20.150 | 27.835 | 10.84 | 48 | 169.00 | 131.77 | 172.33 |
| F9 | 223.75 | 250.58 | 239.61 | 183.97 | 19.06 | 15.61 | 20.347 | 8.097 | 203.1 | 230.54 | 217.6 | 172.2 |
| F10 | 5984.46 | 4211.659 | 4524 | 6108 | 407 | 1432 | 1348 | 369 | 5393 | 1312 | 1425.08 | 5490 |
| F11 | 6931.32 | 6635.03 | 6865 | 6566 | 475.32 | 354.75 | 408.21 | 294.6 | 5868 | 5981 | 6113.02 | 6321 |
| F12 | 2.345 | 2.60 | 2.53 | 2.40 | 0.3794 | 0.2785 | 0.3263 | 0.2383 | 1.7 | 2.221 | 1.746 | 2.017 |
| F13 | 0.5111 | 0.552 | 1.5011 | 0.34246 | 0.1238 | 0.0895 | 0.41817 | 0.0583 | 0.2819 | 0.3974 | 1.083 | 0.2657 |
| F14 | 0.5391 | 0.7987 | 7.732 | 0.28264 | 0.2937 | 0.342 | 4.556 | 0.0343 | 0.29 | 0.28242 | 2.6359 | 0.243 |
| F15 | 17.02 | 23.12 | 66.97 | 15.42 | 2.404 | 10.571 | 77.772 | 1.47 | 14.30 | 17.71 | 27.308 | 12.92 |
| F16 | 12.81 | 12.79 | 12.83 | 11.90 | 0.266 | 0.1874 | 0.1794 | 0.2726 | 12.38 | 12.398 | 12.42 | 11.47 |
| F17 | 965058 | 3981833 | 2540846 | 389902 | 436615 | 2490205 | 1244123 | 326223 | 129986 | 1177515 | 1107044 | 22824 |
| F18 | 199137 | 30841200 | 36432033 | 11283 | 449272 | 80536940 | 34940261 | 9520 | 25671 | 11381 | 7416244 | 2405 |
| F19 | 10.584 | 26.22 | 25.60 | 28.62 | 1.678 | 33.385 | 1.80 | 32.333 | 8.794 | 7.106 | 22.86 | 5.597 |
| F20 | 1463 | 4209.6 | 3737.2 | 348.8 | 807.5 | 2655 | 2537 | 118.5 | 807.2 | 1001.8 | 823.585 | 191.2 |
| F21 | 287610 | 1244033 | 710914 | 153382 | 187045 | 1293203 | 354945 | 121947 | 132730 | 235591 | 299730 | 27539 |
| F22 | 586.5 | 458.5 | 665.70 | 378.992 | 166.88 | 126.3 | 143.81 | 133 | 352.7 | 216.55 | 425 | 173 |
| F23 | 315.244 | 315.257 | 334.029 | 315.244 | 8.78e−13 | 0.0158 | 7.95 | 5.95e−13 | 315.244 | 315.245 | 325.65 | 315.244 |
| F24 | 240.398 | 220.55 | 200.124 | 228.68 | 9.096 | 13.208 | 0.0523 | 7.22 | 222.9 | 201.711 | 200.05 | 222.4 |
| F25 | 205.676 | 212.8 | 217.134 | 204.147 | 3.137 | 4.553 | 4.393 | 0.9164 | 203.06 | 205.69 | 211.18 | 202.84 |
| F26 | 124.44 | 164.57 | 111.935 | 110.34 | 75.72 | 110.71 | 33.7 | 31.53 | 100.27 | 100.501 | 100.61 | 100.27 |
| F27 | 763.611 | 972.56 | 996.85 | 623.0 | 189.926 | 168.99 | 77.46 | 99.14 | 427.3 | 713.631 | 881.77 | 424.97 |
| F28 | 1236.6 | 987.07 | 1282.0 | 1070.2 | 252.42 | 71.279 | 216.6 | 144.7 | 946.5 | 901.36 | 1039.66 | 900.4 |
| F29 | 9508567 | 3573247 | 3049683 | 1965559 | 7769278 | 4629638 | 4599150 | 4156387 | 4780.58 | 1711 | 82249 | 1494 |
| F30 | 22053 | 4434 | 9726 | 4809 | 22730 | 1248 | 7326.9 | 2402 | 2337.4 | 2256 | 4387 | 1934 |
| Nb | 5 | 4 | 1 | 21 | 2 | 3 | 4 | 21 | 5 | 3 | 1 | 23 |

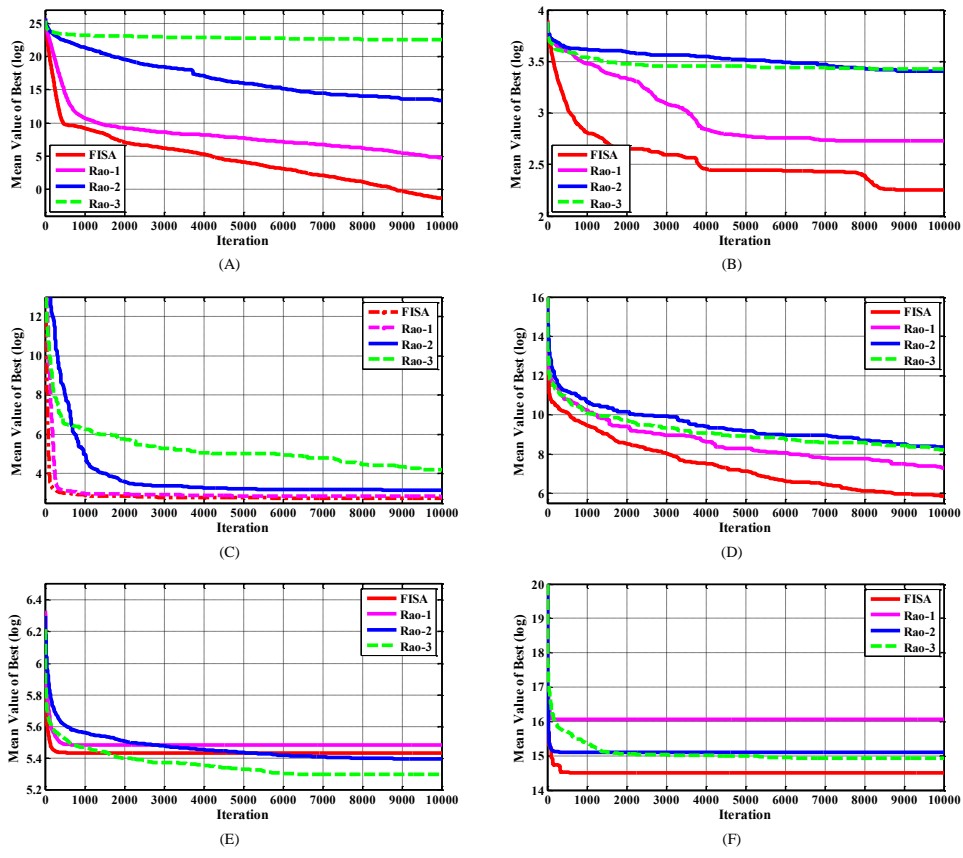

**Figure 3** **The convergence characteristics of FISA and the Rao algorithms for some selected CEC2014 benchmark functions.** (A) F2 test function. (B) F6 test function. (C) F15 test function. (D) F20 test function. (E) F24 test function. (F) F29 test function.

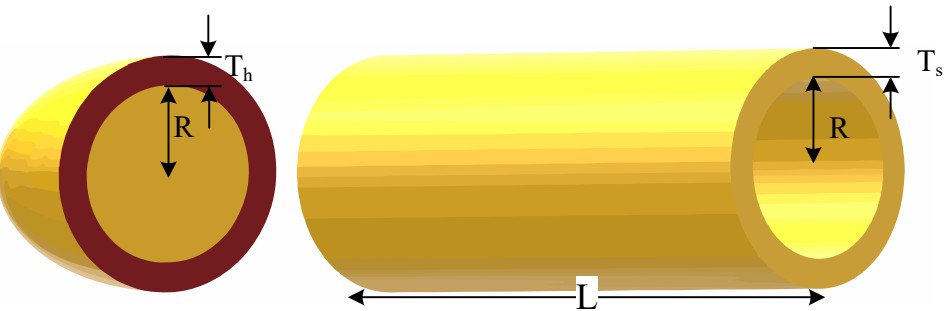

**Figure 4** **Schematic of the pressure vessel design problem.**

and Gaussian QPSO (G-QPSO) (*Coelho, Dos Santos Coelho & Coelho, 2010*), ABC (*Akay & Karaboga, 2012*), a GA equipped with a constraint-handling *via* dominance-based tournament selection (GA4) (*Coello Coello et al., 2002*), co-evolutionary PSO (CPSO) (*He & Wang, 2007*), co-evolutionary DE (CDE) (*Huang, Wang & He, 2007*), unified PSO

**Table 3  Best statistical results of various algorithms for pressure vessel optimal design problem.**

| Methods | Best | Mean | Worst | Std. |
|---|---|---|---|---|
| QPSO (*Coelho, Dos Santos Coelho & Coelho, 2010*) | 6059.7209 | 6440.3786 | 8017.2816 | 479.2671 |
| ABC (*Akay & Karaboga, 2012*) | 6059.714339 | 6245.308144 | N.A. | 2.05e +02 |
| GA4 (*Coello Coello et al., 2002*) | 6059.9463 | 6177.2533 | 6469.3220 | 130.9297 |
| CPSO (*He & Wang, 2007*) | 6061.0777 | 6147.1332 | 6363.8041 | 86.4545 |
| CDE (*Huang, Wang & He, 2007*) | 6059.7340 | 6085.2303 | 6371.0455 | 43.013 |
| G-QPSO (*Coelho, Dos Santos Coelho & Coelho, 2010*) | 6059.7208 | 6440.3786 | 7544.4925 | 448.4711 |
| UPSO (*Parsopoulos & Vrahatis, 2005*) | 6154.70 | 8016.37 | 9387.77 | 745.869 |
| ES (*Mezura-Montes & Coello, 2008*) | 6059.746 | 6850.00 | 7332.87 | 426 |
| T-Cell (*Aragón, Esquivel & Coello, 2010*) | 6390.554 | 6737.065 | 7694.066 | 357 |
| GA3 (*Coello Coello, 2000*) | 6288.7445 | 6293.8432 | 6308.4970 | 7.4133 |
| HAIS-GA (*Bernardino et al., 2008*) | 6832.584 | 7187.314 | 8012.615 | 276 |
| CSA (*Askarzadeh, 2016*) | 6059.71436343 | 6342.49910551 | 7332.84162110 | 384.94541634 |
| BFOA (*Mezura-Montes & Hernández-Ocana, 2008*) | 6060.460 | 6074.625 | N.A. | 156 |
| BIANCA (*Montemurro, Vincenti & Vannucci, 2013*) | 6059.9384 | 6182.0022 | 6447.3251 | 122.3256 |
| QS (*Zhang et al., 2018*) | 6059.714 | 6060.947 | 6090.526 | N.A. |
| Rao-1 (*Rao & Pawar, 2020b*) | 6059.714334 | 6069.230694 | 6093.903548 | 10.451664 |
| Rao-2 (*Rao & Pawar, 2020b*) | 6059.714334 | 6062.055668 | 6090.526202 | 7.171409 |
| Rao-3 (*Rao & Pawar, 2020b*) | 6059.714334 | 6061.883052 | 6090.526202 | 7.810982 |
| FISA | 6059.714334 | 6061.320721 | 6066.824063 | 4.74 |

(UPSO) (*Parsopoulos & Vrahatis, 2005*), Crow search algorithm (CSA) (*Askarzadeh, 2016*), hybridizing a genetic algorithm with an artificial immune system (HAIS-GA) (*Bernardino et al., 2008*), bacterial foraging optimization algorithm (BFOA) (*Mezura-Montes & Hernández-Ocana, 2008*), evolution strategies (ES) (*Mezura-Montes & Coello, 2008*), A modification of the T-Cell algorithm (*Aragón, Esquivel & Coello, 2010*), a GA enhanced with a self-adaptive penalty method (GA3) (*Coello Coello, 2000*), queuing search (QS) algorithm (*Zhang et al., 2018*), and a GA equipped with a constraint-handling *via* automatic dynamic penalization (ADP) method (BIANCA) (*Montemurro, Vincenti & Vannucci, 2013*). Furthermore, the best solution found for the problem using the suggested technique was shown in Table 4. Tables 3 and 4 demonstrate that FISA outperforms other algorithms with the smallest value for the best solution, and ranking first for the worst solution and average value. This confirms that FISA performs well and reliably in tackling the pressure vessel optimal design problem.

### Tension/compression spring optimal design

According to Fig. 5, the goal of the problem includes reducing the tension/compression spring weight while satisfying four inequality limitations (one linear and three nonlinear). The problem comprises three continuous design variables: wire diameter (denoted as $d$ or $x_1$), mean coil diameter (denoted as $D$ or $x_2$), and the number of active coils (denoted as $P$ or $x_3$) (*Askarzadeh, 2016*).

Minimize:

$$f(X) = (x_3 + 2)x_2 x_1^2 \tag{13}$$

**Table 4** **The best solutions found for the pressure vessel optimal design problem by FISA.**

| Design variables | FISA |
|---|---|
| $x_1$ | 0.8125 |
| $x_2$ | 0.4375 |
| $x_3$ | 42.098446 |
| $x_4$ | 176.63660 |
| $g_1(X)$ | $-1.1300e{-}10$ |
| $g_2(X)$ | $-0.035881$ |
| $g_3(X)$ | $-2.788752e{-}05$ |
| $g_4(X)$ | $-63.36340$ |
| Best | 6059.714334 |

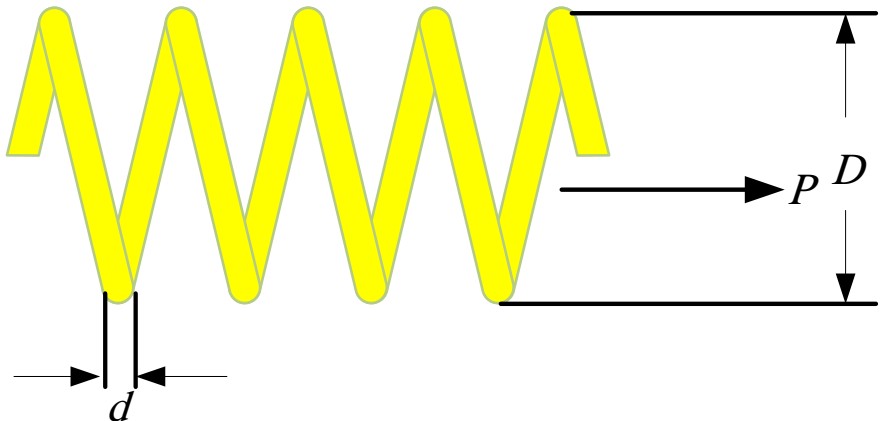

**Figure 5** **The tension/compression spring optimal design problem.**

subject to:

$$g_1(X) = 1 - \frac{x_2^3 x_3}{71,785 x_1^4} \leq 0, \tag{14}$$

$$g_2(X) = \frac{4x_2^2 - x_1 x_2}{12,566\left(x_1^3 x_2 - x_1^4\right)} + \frac{1}{5,108 x_1^2} - 1 \leq 0, \tag{15}$$

$$g_3(X) = 1 - \frac{140.45 x_1}{x_2^2 x_3} \leq 0, \tag{16}$$

$$g_4(X) = \frac{x_1 + x_2}{1.5} - 1 \leq 0. \tag{17}$$

$0.05 \leq x_1 \leq 2$, $0.25 \leq x_2 \leq 1.3$, $2 \leq x_3 \leq 15$.

Table 5 describes the outcomes of the proposed algorithm for the problem in comparison with other standard and well-known algorithms including BFOA (*Mezura-Montes &*

**Table 5  Best statistical results of various algorithms for tension/compression spring optimal design problem.**

| Methods | Best | Mean | Worst | Std. |
|---|---|---|---|---|
| CA (*Coello Coello & Becerra, 2004*) | 0.012721 | 0.013568 | 0.0151156 | 8.4e−04 |
| BFOA (*Mezura-Montes & Hernández-Ocana, 2008*) | 0.012671 | 0.012759 | N.A. | 1.36e−04 |
| T-Cell (*Aragón, Esquivel & Coello, 2010*) | 0.012665 | 0.012732 | 0.013309 | 9.4e−05 |
| CDE (*Huang, Wang & He, 2007*) | 0.012670 | 0.012703 | 0.012790 | 2.07e−05 |
| CPSO (*He & Wang, 2007*) | 0.0126747 | 0.012730 | 0.012924 | 5.19e−05 |
| TEO (*Kaveh & Dadras, 2017*) | 0.012665 | 0.012685 | 0.012715 | 4.4079e−06 |
| G-QPSO (*Coelho, Dos Santos Coelho & Coelho, 2010*) | 0.012665 | 0.013524 | 0.017759 | 1.268e−03 |
| SBO (*Ray & Liew, 2003*) | 0.012669249 | 0.012922669 | 0.016717272 | 5.92e−04 |
| GA4 (*Coello Coello et al., 2002*) | 0.012681 | 0.012742 | 0.012973 | 9.5e−05 |
| GA3 (*Coello Coello, 2000*) | 0.0127048 | 0.012769 | 0.012822 | 3.93e−05 |
| (l + k)-ES (*Mezura-Montes & Coello, 2005*) | 0.012689 | 0.013165 | N.A. | 3.9e−04 |
| UPSO (*Parsopoulos & Vrahatis, 2005*) | 0.01312 | 0.02294 | N.A. | 7.2e−03 |
| GWO (*Mirjalili, Mirjalili & Lewis, 2014*) | 0.0126660 | N.A. | N.A. | N.A. |
| WCA (*Eskandar et al., 2012*) | 0.012665 | 0.012746 | 0.012952 | 8.06e−05 |
| BIANCA (*Montemurro, Vincenti & Vannucci, 2013*) | 0.012671 | 0.012681 | 0.012913 | 5.1232e−05 |
| SDO (*Zhao, Wang & Zhang, 2019*) | 0.0126663 | 0.0126724 | 0.0126828 | 6.1899e−06 |
| QS (*Zhang et al., 2018*) | 0.012665 | 0.012666 | 0.012669 | N.A. |
| Rao-1 (*Rao & Pawar, 2020b*) | 0.012666 | 0.012712 | 0.012846 | 3.6195e−05 |
| Rao-2 (*Rao & Pawar, 2020b*) | 0.012669 | 0.013232 | 0.030455 | 2.5886e−03 |
| Rao-3 (*Rao & Pawar, 2020b*) | 0.012672 | 0.013086 | 0.017773 | 1.2062e−03 |
| FISA | 0.012665 | 0.012666 | 0.012675 | 7.05e−07 |

*Hernández-Ocana, 2008*), T-Cell (*Aragón, Esquivel & Coello, 2010*), CDE (*Huang, Wang & He, 2007*), a cultural algorithm (CA) (*Coello Coello & Becerra, 2004*), CPSO (*He & Wang, 2007*), GA4 (*Coello Coello et al., 2002*), GA3 (*Coello Coello, 2000*), TEO (*Kaveh & Dadras, 2017*), G-QPSO (*Coelho, Dos Santos Coelho & Coelho, 2010*), SBO (*Ray & Liew, 2003*), evolutionary algorithms ((l + k)-ES) (*Mezura-Montes & Coello, 2005*), grey wolf optimizer (GWO) (*Mirjalili, Mirjalili & Lewis, 2014*), UPSO (*Parsopoulos & Vrahatis, 2005*), QS (*Zhang et al., 2018*), SDO (*Zhao, Wang & Zhang, 2019*), BIANCA (*Montemurro, Vincenti & Vannucci, 2013*), and water cycle algorithm (WCA) (*Eskandar et al., 2012*). Furthermore, Table 6 shows the optimal solution obtained for the problem using the suggested algorithm. The tables demonstrate that FISA outperforms all other algorithms in terms of the best value, with the worst solution and average value being the smallest. This suggests that FISA is more effective than other competitive optimizers in solving this problem.

## Welded beam optimal design

The goal of the problem is the minimization of the cost of a welded beam. The problem has four continuous decision variables, as depicted in Fig. 6, namely $x_1(h), x_2(l), x_3(t)$, and $x_4(b)$, along with two linear and five nonlinear inequality limitations (*Askarzadeh, 2016*). Minimize:

$$f(X) = 1.10471 x_2 x_1^2 + 0.04811 x_3 x_4 (14 + x_2) \tag{18}$$

**Table 6** The best solutions found for the tension/compression spring optimal design problem by FISA.

| Design variables | FISA |
| --- | --- |
| $x_1$ | 0.0517770562 |
| $x_2$ | 0.3588357559 |
| $x_3$ | 11.1661043232 |
| $g_1(X)$ | $-1.310475e{-05}$ |
| $g_2(X)$ | $-5.853421e{-06}$ |
| $g_3(X)$ | $-4.057851$ |
| $g_4(X)$ | $-0.727695$ |
| Best | 0.012665 |

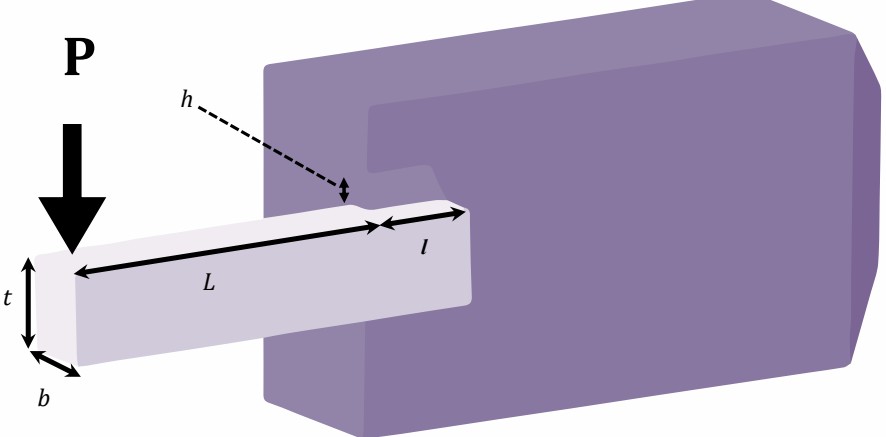

**Figure 6** Schematic of welded beam optimal design problem.

Subject to:

$$g_1(X) = \tau(x) - \tau_{\max} \leq 0, \tag{19}$$

$$g_2(X) = \sigma(x) - \sigma_{\max} \leq 0, \tag{20}$$

$$g_3(X) = x_1 - x_4 \leq 0, \tag{21}$$

$$g_4(X) = 0.10471x_1^2 + 0.04811x_3x_4(14 + x_2) - 5 \leq 0. \tag{22}$$

$$g_5(X) = 0.125 - x_1 \leq 0, \tag{23}$$

$$g_6(X) = \delta(x) - \delta_{\max} \leq 0, \tag{24}$$

$$g_7(X) = P - P_c(x) \leq 0, \tag{25}$$

Where

$$\tau(x) = \sqrt{(\tau')^2 + 2\tau'\tau''\frac{x_2}{2R} + (\tau'')^2} \tag{26}$$

$$\tau' = \frac{P}{\sqrt{2}x_1 x_2}, \qquad \tau'' = \frac{MR}{J}, \tag{27-28}$$

$$M = P\left(L + \frac{x_2}{2}\right), \qquad R = \sqrt{\frac{x_2^2}{4} + \left(\frac{x_1 + x_3}{2}\right)^2}, \qquad \delta(x) = \frac{4PL^3}{Ex_3^3 x_4} \tag{29-31}$$

$$J = 2\left[\sqrt{2}x_1 x_2\left\{\frac{x_2^2}{12} + \left(\frac{x_1 + x_3}{2}\right)^2\right\}\right], \qquad \sigma(x) = \frac{6PL}{x_4 x_3^2}, \tag{32-33}$$

$$P_c(x) = \frac{4.013E\sqrt{\frac{x_4^6 x_3^2}{36}}}{L^2}\left(1 - \frac{x_3}{2L}\sqrt{\frac{E}{4G}}\right), \tag{34}$$

$P = 6,000$ lb; $L = 14$ in; $E = 30e6$ psi $G = 12e6$ psi, $\tau_{max} = 13,000$ psi, $\sigma_{max} = 30,000$ psi $\delta_{max} = 0.25$ in, $0.1 \leq x_1 \leq 2$, $0.1 \leq x_2 \leq 10$, $0.1 \leq x_3 \leq 10$, $0.1 \leq x_4 \leq 2$.

The performance of the suggested method for the problem is presented in Table 7, where it is compared to other widely used algorithms, including a cooperative PSO with stochastic movements (EPSO) (*Ngo, Sadollah & Kim, 2016*), BFOA (*Mezura-Montes & Hernández-Ocana, 2008*), T-Cell (*Aragón, Esquivel & Coello, 2010*), CDE (*Huang, Wang & He, 2007*), a hybrid real-parameter GA (HSA-GA) (*Hwang & He, 2006*), CPSO (*He & Wang, 2007*),TEO (*Kaveh & Dadras, 2017*), SBO (*Ray & Liew, 2003*), Derivative-Free Filter Simulated Annealing Method (FSA) (*Hedar & Fukushima, 2006*), GA4 (*Coello Coello et al., 2002*), (l + k)-ES (*Mezura-Montes & Coello, 2005*), GWO (*Mirjalili, Mirjalili & Lewis, 2014*), SFO (*Shadravan, Naji & Bardsiri, 2019*), HGSO (*Hashim et al., 2019*), UPSO (*Parsopoulos & Vrahatis, 2005*), WCA (*Eskandar et al., 2012*), BIANCA (*Montemurro, Vincenti & Vannucci, 2013*). Furthermore, Table 8 displays the optimal solution obtained by the suggested algorithm for the problem. Table 7 and 8 demonstrate that FISA is capable of discovering the most effective optimization solution. The statistical findings also indicate that FISA surpasses other methods and can more effectively handle the constrained engineering problems.

## MANAGERIAL IMPLICATIONS

Metaheuristic methods provide managers and decision makers with reliable tools for finding appropriate solutions to real-world problems with a limited computational burden and a limited time. Since there are major difficulties in finding the exact solutions of a wide

**Table 7   Best statistical results of various algorithms for welded beam optimal design problem.**

| Methods | Best | Mean | Worst | Std. |
|---|---|---|---|---|
| EPSO (*Ngo, Sadollah & Kim, 2016*) | 1.7248530 | 1.7282190 | 1.7472200 | 5.62e−03 |
| BFOA (*Mezura-Montes & Hernández-Ocana, 2008*) | 2.3868 | 2.4040 | N.A. | 1.6e−02 |
| T-Cell (*Aragón, Esquivel & Coello, 2010*) | 2.3811 | 2.4398 | 2.7104 | 9.314e−02 |
| CDE (*Huang, Wang & He, 2007*) | 1.73346 | 1.768158 | 1.824105 | 2.2194e−02 |
| CPSO (*He & Wang, 2007*) | 1.728024 | 1.748831 | 1.782143 | 1.2926e−02 |
| HSA-GA (*Hwang & He, 2006*) | 2.2500 | 2.26 | 2.28 | 7.8e−03 |
| FSA (*Hedar & Fukushima, 2006*) | 2.3811 | 2.4041 | 2.4889 | N.A. |
| TEO (*Kaveh & Dadras, 2017*) | 1.725284 | 1.768040 | 1.931161 | 5.81661e−02 |
| SBO (*Ray & Liew, 2003*) | 2.3854347 | 3.0025883 | 6.3996785 | 9.59e−01 |
| GA4 (*Coello Coello et al., 2002*) | 1.728226 | 1.792654 | 1.993408 | 7.47e−02 |
| (l + k)-ES (*Mezura-Montes & Coello, 2005*) | 1.724852 | 1.777692 | N.A. | 8.8e−02 |
| UPSO (*Parsopoulos & Vrahatis, 2005*) | 1.92199 | 2.83721 | N.A. | 6.83e−01 |
| GWO (*Mirjalili, Mirjalili & Lewis, 2014*) | 1.72624 | N.A. | N.A. | N.A. |
| SFO (*Shadravan, Naji & Bardsiri, 2019*) | 1.73231 | N.A. | N.A. | N.A. |
| HGSO (*Hashim et al., 2019*) | 1.7260 | 1.7265 | 1.7325 | 7.66e−03 |
| WCA (*Eskandar et al., 2012*) | 1.724856 | 1.726427 | 1.744697 | 4.29e−03 |
| BIANCA (*Montemurro, Vincenti & Vannucci, 2013*) | 1.725436 | 1.752201 | 1.793233 | 2.3001e−02 |
| Rao-1 (*Rao & Pawar, 2020b*) | 1.724852 | 1.724852 | 1.724852 | 2.62 |
| Rao-2 (*Rao & Pawar, 2020b*) | 1.724852 | 1.724852 | 1.724852 | 9.83e−04 |
| Rao-3 (*Rao & Pawar, 2020b*) | 1.724852 | 1.724852 | 1.724852 | 2.06e−03 |
| FISA | 1.724852 | 1.724852 | 1.724852 | 5.93e−05 |

**Table 8   The best solutions found for the welded beam optimal design problem by FISA.**

| Design variables | FISA |
|---|---|
| $x_1$ | 0.20572963980 |
| $x_2$ | 3.4704886655 |
| $x_3$ | 9.0366239101 |
| $x_4$ | 0.2057296398 |
| $g_1(X)$ | −2.2653330e−07 |
| $g_2(X)$ | −3.1932722e−07 |
| $g_3(X)$ | 0.0 |
| $g_4(X)$ | −3.43298379 |
| $g_5(X)$ | −0.08072964 |
| $g_6(X)$ | −0.23554032 |
| $g_7(X)$ | −1.10549263e−06 |
| Best | 1.7248523 |

variety of real-world problems, metaheuristic methods are still the focus of many studies for tackling these issues. The article proposed a simple non-parametric algorithm, named Fully Informed Search Algorithm. The suggested algorithm's effectiveness was verified by testing it on both shifted benchmark functions and mechanical design problems. The non-parametric nature of the proposed method along with its good performance in

finding high quality solution of complicated real-world optimization problems make it a good choice for supporting managers in decision making without having to deal with sophisticated parameter tuning.

## CONCLUSIONS

In this article, a new and powerful variant of Rao algorithms, entitled Fully Informed Search Algorithm (FISA), is suggested to enhance the performance of Rao algorithms in optimizing real-parameter shifted functions. The efficacy of the suggested algorithm was assessed compared to three original Rao algorithms for test functions presented in CEC 2005 and CEC 2014 and engineering design optimization problems. The obtained results demonstrated that the proposed algorithm has a much better performance compared to the original Rao algorithms.

On the other hand, each algorithm has its own limitations. For this reason, after presenting each algorithm, its improved and modified versions are published one after another in different formats and forms. Like any other algorithm, the proposed algorithm can have limitations, like low convergence speed or getting stuck in local optima. Here we propose several ways to improve and evolve this algorithm. However, the most effective way to evaluate the performance of an algorithm for any given problem is through experimental testing.

In future studies, the suggested FISA can be utilized for solving various complex optimization problems that occur in the real world. Additionally, our future plans involve creating binary and multiobjective variants of FISA. Also, the optimization of support vector machines or kernel extreme learning machines is possible using FISA. By merging FISA with other algorithms, we can establish new hybrid algorithms that take advantage of the strengths and abilities of both algorithms. Recently, many studies in high-dimensional optimization have been conducted, with the majority focusing on the cooperative co-evolution technique. In upcoming studies, FISA could be integrated into various cooperative co-evolution frameworks with different classes to enhance its effectiveness. Additionally, FISA can tackle other practical large-scale optimization problems. Moreover, initializing with opposite learning in FISA would be an appropriate domain to be explored in the future.

### Funding

The research was supported by the Project of Excellence of Faculty of Science, University of Hradec Kralove, Czech Republic, No. 2210/2023-2024. The funders had no role in study design, data collection and analysis, decision to publish, or preparation of the manuscript.

### Grant Disclosures

The following grant information was disclosed by the authors:
Project of Excellence of Faculty of Science, University of Hradec Kralove, Czech Republic: No. 2210/2023-2024.

## Competing Interests

The authors declare there are no competing interests.

## Author Contributions

- Mojtaba Ghasemi performed the computation work, prepared figures and/or tables, authored or reviewed drafts of the article, and approved the final draft.
- Abolfazl Rahimnejad conceived and designed the experiments, performed the experiments, analyzed the data, authored or reviewed drafts of the article, and approved the final draft.
- Ebrahim Akbari analyzed the data, performed the computation work, prepared figures and/or tables, authored or reviewed drafts of the article, and approved the final draft.
- Ravipudi Venkata Rao conceived and designed the experiments, performed the experiments, analyzed the data, authored or reviewed drafts of the article, and approved the final draft.
- Pavel Trojovský conceived and designed the experiments, performed the experiments, analyzed the data, authored or reviewed drafts of the article, and approved the final draft.
- Eva Trojovská conceived and designed the experiments, performed the experiments, analyzed the data, authored or reviewed drafts of the article, and approved the final draft.
- Stephen Andrew Gadsden analyzed the data, authored or reviewed drafts of the article, and approved the final draft.

## Data Availability

The MATLAB Codes of the proposed algorithm (Fully Informed Search Algorithm) are available as a Supplemental File.

The data for the CEC 2005 and CEC 2014 benchmark functions are available at GitHub: https://github.com/P-N-Suganthan/CEC2005 and https://github.com/P-N-Suganthan/CEC2014.

The data is part of the repository of Professor Ponnuthurai Nagaratnam Suganthan who is a computer science academic at the Nanyang Technological University, Singapore.

## Supplemental Information

Supplemental information for this article can be found online at http://dx.doi.org/10.7717/peerj-cs.1431#supplemental-information.

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
