# Peer review of "A new metaphor-less simple algorithm based on Rao algorithms: a Fully Informed Search Algorithm (FISA)"

_PeerJ Computer Science, doi:10.7717/peerj-cs.1431_

## Round 0.1 · original submission · Major Revisions

The paper must be significantly improved. Please follow the reviewers' suggestions.

Reviewer 1 ·

Basic reporting

The authors propose a new version of the Rao nonparametric algorithm and named it the Fully Informed Search Algorithm (FISA). The article includes an appropriate introduction and background to demonstrate how the work fits into the broader field of knowledge. However, I suggest the authors check the recent literature sources, from the previous two, or three years, to make the article up-to-date.
The structure of the article is in accordance with a standard format: Introduction, Related work, Methods, Results, and Conclusions. Figures are relevant to the content of the article, appropriately described and labeled. The paper includes all results relevant to the hypothesis.

Experimental design

The research question is well-defined, relevant and meaningful. It is explained how the research fills an identified knowledge gap.
The research is conducted in conformity with the prevailing ethical standards in the field. Methods are described with sufficient detail.

Validity of the findings

The obtained results are of interest to the general and academic audience. The conclusions are appropriately stated and connected to the original investigated question.
I suggest the following improvements:
1. It should be further discussed about future directions of research.
2. What are the limitations of this study?
3. It should be further discussed about the managerial implications of the research.

Reviewer 2 ·

Basic reporting

I propose to indicate more clearly in the title the broader application of the model - real-world optimization problems or more specifically - optimizing real-parameter shifted functions.
The summary and conclusion correspond to the essence of the work.
The use of terminology is correct and it complies with applicable standards.

Experimental design

The authors clearly explained the methodology and structure of the work. The authors clearly presented a well-known optimization problem, and ways and methods of solving it. The authors presented the limitations of metaheuristic models in solving optimization problems and indicated the need to develop new ones with special reference to the RAO algorithms.
In line, 177 authors said, „These functions have been successfully utilized in many articles“. I suggest the authors add appropriate references.
In the paper, the results for selected engineering problems are presented in a good way, tabularly, and clearly. I recommend that that part be supplemented with a short comment, or analysis, of the obtained results: line 251(Table 3, Table 4), line 280 (Table 5, Table 6), and line 321 (Table 7, Table 8).

Validity of the findings

The effectiveness of the proposed algorithm was assessed in comparison with three original Rao algorithms for test functions presented in CEC 317 2005 and CEC 2014 and engineering design optimization problems.
By comparing the results of the new model with the existing models, the justification of the application can be seen, and the authors plan further application of the model in other optimization problems in future research.

---

## Round 0.2 · accepted · Accept

All reviewers' comments have been addressed. The paper can be accepted.

Reviewer 1 ·

Basic reporting

The revised manuscript has been properly improved.

Experimental design

The revised manuscript has been properly improved.

Validity of the findings

The revised manuscript has been properly improved.

Reviewer 2 ·

Basic reporting

no comment

Experimental design

no comment

Validity of the findings

no comment

Additional comments

The authors supplemented the work in an adequate way and acted according to all suggestions